# iWGAN: AN AUTOENCODER WGAN FOR INFERENCE

## ABSTRACT

Generative Adversarial Networks (GANs) have been impactful on many problems and applications but suffer from unstable training. Wasserstein GAN (WGAN) leverages the Wasserstein distance to avoid the caveats in the minmax two-player training of GANs but has other defects such as mode collapse and lack of metric to detect the convergence. We introduce a novel inference WGAN (iWGAN) model, which is a principled framework to fuse auto-encoders and WGANs. The iWGAN jointly learns an encoder network and a generative network using an iterative primal dual optimization process. We establish the generalization error bound of iWGANs. We further provide a rigorous probabilistic interpretation of our model under the framework of maximum likelihood estimation. The iWGAN, with a clear stopping criteria, has many advantages over other autoencoder GANs. The empirical experiments show that our model greatly mitigates the symptom of mode collapse, speeds up the convergence, and is able to provide a measurement of quality check for each individual sample. We illustrate the ability of iWGANs by obtaining a competitive and stable performance with state-of-the-art for benchmark datasets.

## 1 INTRODUCTION

One of the goals of generative modeling is to match the model distribution $P_\theta(x)$ with parameters $\theta$ to the true data distribution $P_X$ for a random variable $X \in \mathcal{X}$. For latent variable models, the data point $X$ is generated from a latent variable $Z \in \mathcal{Z}$ through a conditional distribution $P(X|Z)$. Here $\mathcal{X}$ denotes the support for $P_X$ and $\mathcal{Z}$ denotes the support for $P_Z$. There has been a surge of research on deep generative networks in recent years and the literature is too vast to summarize here (Kingma & Welling, 2013; Goodfellow et al., 2014; Li et al., 2015). These models have provided a powerful framework for modeling complex high dimensional datasets. We start introducing two main approaches for generative modeling. The first one is called *variational auto-encoders* (VAEs) (Kingma & Welling, 2013), which use variational inference to learn a model by maximizing the lower bound of the likelihood function. VAEs have elegant theoretical foundations but the drawback is that they tend to produce blurry images. The second approach is called *generative adversarial networks* (GANs) (Goodfellow et al., 2014), which learn a model by using a powerful discriminator to distinguish between real data points and generative data points. GANs produce more visually realistic images but suffer from the unstable training and the mode collapse problem. Although there are many variants of generative models trying to take advantages of both VAEs and GANs (Tolstikhin et al., 2017; Rosca et al., 2017), to the best of our knowledge, the model which provides a unifying framework combining the best of VAEs and GANs in a principled way is yet to be discovered.

### 1.1 RELATED WORK

**GANs and WGANs:** The generative model is to learn a mapping, denoted by $G$, from $\mathcal{Z}$ to $\mathcal{X}$ to approximate the conditional distribution $P_G(X|Z)$ of the data point $X \in \mathcal{X}$ given latent code $Z \in \mathcal{Z}$. We consider the deterministic mapping $G$ in this work. Both GANs and Wasserstein GANs (WGANs) (Arjovsky et al., 2017) can be viewed as minimizing certain divergence between the data distribution $P_X$ and the generative model distribution $P_{G(Z)}$. For example, the Jensen-Shannon (JS) divergence is implicitly used in GANs (Goodfellow et al., 2014). The 1-Wasserstein distance between $P_X$ and $P_{G(Z)}$, denoted by $W_1(P_X, P_{G(Z)})$, is employed in WGANs. Empirical experiments suggest that the Wasserstein distance is a more sensible measure to differentiate probability measures supported in low-dimensional manifold. In terms of training, it turns out that it is hard or even impossible to

compute these standard divergences in probability, especially when $P_X$ is unknown and $P_{G(Z)}$ is parameterized by deep neural networks (DNNs). The training of GANs is converted into playing a game between two competing networks: the *generator* and the *discriminator*. The generator is to fool the discriminator and the discriminator is to distinguish between true data samples and generated samples. Instead, the training of WGANs is to study its dual problem because of the elegant form of Kantorovich-Rubinstein duality (Villani, 2008). Analogous to GANs, the discriminator is now a real-valued 1-Lipschitz function. Many techniques such as weight clipping (Arjovsky et al., 2017) and gradient penalty (Gulrajani et al., 2017) are used to enforce the Lipschitz constraint. More discussions about WGANs will be presented in Section 2.

**Autoencoder GANs:** Larsen et al. (2016) first introduced the VAE-GAN, which is a hybrid of VAEs and GANs. The VAE-GAN uses a GAN discriminator to replace a VAE's decoder to learn the loss function. The motivation behind this modification is that VAEs tend to produce blurry outputs during the reconstruction phase. More recent VAE-GAN variants, such as Adversarial Generator Encoders (AGE) (Ulyanov et al., 2018) and Auto-encoding GANs ($\alpha$-GAN) (Rosca et al., 2017), use a separate encoder to stabilize GAN training. The main difference with standard GANs is that, besides the generator $G$, there is an encoder $Q : \mathcal{X} \to \mathcal{Z}$ which maps the data points into the latent space. This encoder is to approximate the conditional distribution $Q(Z|X)$ of the latent variable $Z$ given the data point $X$. Other encoder-decoder GANs are introduced in Adversarially Learned Inference (ALI) (Dumoulin et al., 2016) and Bidirectional Generative Adversarial Networks (BiGAN) (Donahue et al., 2016). The objective of both ALI and BiGAN is to match two joint distributions under the framework of vanilla GANs, the joint distribution of $(X, Q(X))$ and the joint distribution of $(G(Z), Z)$. When the algorithm achieves equilibrium, these two joint distributions roughly match. We are able to obtain more meaningful latent codes by $Q(X)$, and this should improve the quality of the generator as well. Adversarial Variational Bayes (AVB) (Mescheder et al., 2017) presented a more flexible latent distribution to train Variational Autoencoders. Hu et al. (2017) provided new interpretations of GANs and VAEs and revealed strong connections between them which are linked by the classic wake-sleep algorithm.

**Duality in GANs:** Regarding the optimization perspectives of GANs, (Chen et al., 2018; Zhao et al., 2018) studied duality-based methods for improving algorithm performance for training. Primal-dual Wasserstein GANs (PD-GANs) are introduced in (Gemici et al., 2018), which proposed a new penalty term whose evaluation samples are obtained from the encoder $Q$. Farnia & Tse (2018) developed a convex duality framework to address the case when the discriminator is constrained into a smaller class. Grnarova et al. (2018) developed an evaluation metric to detect the non-convergence behavior of vanilla GANs, which is the duality gap defined as the difference between the primal and the dual objective functions. Husain et al. (2019) investigated the close relationship between WAE (Tolstikhin et al., 2017) and f-GANs (Nowozin et al., 2016), and proved generalization results for autoencoder models.

## 1.2 Our Contributions

Although there are many interesting works on autoencoder GANs, it remains unclear what the principles are underlying the fusion of auto-encoders and GANs. For example, do there even exist these two mappings, the encoder $Q$ and the decoder $G$, for any high-dimensional random variable $X$, such that $Q(X)$ has the same distribution as $Z$ and $G(Z)$ has the same distribution as $X$? Is there any probabilistic interpretation such as the maximum likelihood principle on autoencoder GANs? We introduce inference Wasserstein GANs (iWGANs), which provide satisfying answers for these questions. We focus on the 1-Wasserstein distance, instead of the Kullback-Leibler divergence. We borrow the strength from both the primal and the dual problems and demonstrate the synergistic effect between these two optimizations. The encoder component tends out to be a natural consequence from our algorithm. Furthermore, the iWGAN has a rigorous probabilistic interpretation under the maximum likelihood principle, and our learning algorithm is equivalent to the maximum likelihood estimation when our model is defined as an energy-based model based on an autoencoder. Our main contributions are listed as below:

1. We propose a novel framework, called iWGAN, to learn both an encoder and a decoder simultaneously. We prove the existence of meaningful encoder and decoder, establish an equivalence between WGAN and iWGAN, and develop a generalization error bound for iWGAN.

2. We establish a rigorous probability interpretation for iWGANs and our training process is exactly the same as the maximum likelihood estimation. As a byproduct, this interpretation allows us to perform the quality check at the individual sample level.

3. We demonstrate the natural use of the duality gap as a measure of convergence for iWGANs, and show its effectiveness for various numerical settings. Our experiments do not experience any mode collapse problem.

## 2   iWGAN

The autoencoder generative model consists of two parts: an encoder $Q$ and a generator $G$. The encoder $Q$ maps a data sample $x \in \mathcal{X}$ to a latent variable $z \in \mathcal{Z}$, and the generator $G$ takes a latent variable $z \in \mathcal{Z}$ to produce a sample $G(z)$. The autoencoder generative model should satisfy the following three conditions *simultaneously*: (a) The generator can generate images which have a similar distribution with observed images; (b) The encoder can produce meaningful encodings in the latent space; (c) The reconstruction errors of this model based on these meaningful encodings are small. The benefit of using an autoencoder is to encourage the model to better represent *all* the data it is trained with, so that it discourages mode-collapse.

We first show that, for any distribution residing on a smooth manifold, there always exists an encoder $Q^*$ which guarantees meaningful encodings and exists a generator $G^*$ which generates samples with the same distribution as data points by using these meaningful codes.

**Theorem 2.1.** *Consider a continuous random variable $X \in \mathcal{X}$, where $\mathcal{X}$ is a $d$-dimensional smooth Riemannian manifold. Then, there exist two mappings $Q^* : \mathcal{X} \to \mathbb{R}^p$ and $G^* : \mathbb{R}^p \to \mathcal{X}$, with $p = \max\{d(d+5)/2, d(d+3)/2 + 5\}$, such that $Q^*(X)$ follows a multivariate normal distribution with zero mean and identity covariance matrix and $G^* \circ Q^*$ is an identity mapping, i.e., $X = G^*(Q^*(X))$.*

Learning $Q^*$ and $G^*$ from the data points is a challenging task. Recall that the 1-Wasserstein distance between the data distribution $P_X$ and the generative model distribution $P_{G(Z)}$ is defined as

$$W_1(P_X, P_{G(Z)}) = \inf_{\pi \in \Pi(P_X, P_Z)} \mathbb{E}_{(X,Z) \sim \pi} \|X - G(Z)\|, \tag{1}$$

where $\|\cdot\|$ represents the $L_2$-norm and $\Pi(P_X, P_Z)$ is a set of all joint distributions of $(X, Z)$ with marginal measures $P_X$ and $P_Z$, respectively. The main difficulty in (1) is to find the optimal coupling $\pi$ and this is a constrained optimization with $P_X(x) = \int \pi(x, z) dz$ for $x \in \mathcal{X}$ and $P_Z(z) = \int \pi(x, z) dx$ for $z \in \mathcal{Z}$.

Based on the Kantorovich-Rubinstein duality, the WGAN studies the 1-Wasserstein distance (1) from the dual format

$$W_1(P_X, P_{G(Z)}) = \sup_{f \in \mathcal{F}} \left\{ \mathbb{E}_{X \sim P_X} \left[ f(X) \right] - \mathbb{E}_{Z \sim P_Z} \left[ f(G(Z)) \right] \right\}, \tag{2}$$

where $\mathcal{F}$ is the set of all bounded 1-Lipschitz functions. Weight clipping (Arjovsky et al., 2017) and gradient penalty (Gulrajani et al., 2017) have been used to satisfy the constraint of Lipschitz continuity. The experiment of (Arjovsky et al., 2017) showed that the WGAN can avoid the problem of gradient vanishment. However, the WGAN does not produce meaningful encodings and many experiments still display the problem of mode collapse (Arjovsky et al., 2017; Gulrajani et al., 2017). On the other hand, the Wasserstein Autoencoder (WAE) (Tolstikhin et al., 2017), after introducing an encoder $Q : \mathcal{X} \to \mathcal{Z}$ to approximate the conditional distribution of $Z$ given $X$, studies the reconstruction error $\inf_{Q \in \mathcal{Q}} \mathbb{E}_X \|X - G(Q(X))\|$, where $\mathcal{Q}$ is a set of encoder mappings whose elements satisfies $P_{Q(X)} = P_Z$. The penalty, such as $\mathcal{D}(P_{Q(X)}, P_Z)$, is added to the objective to satisfy this constraint, where $\mathcal{D}$ is an arbitrary divergence between $P_{Q(X)}$ and $P_Z$. The WAE can produce meaningful encodings and controlled reconstruction error. However, the WAE defines a generative model in an implicit way and does not model the generator through $G(Z)$ with $Z \sim P_Z$ directly.

To take the advantages of both WGAN and WAE, we propose a new autoencoder GAN model, called iWGAN, whose objective is

$$\overline{W}_1(P_X, P_{G(Z)}) = \inf_{Q \in \mathcal{Q}} \sup_{f \in \mathcal{F}} \mathbb{E}_X \|X - G(Q(X))\| + \mathbb{E}_X \left[ f(G(Q(X))) \right] - \mathbb{E}_Z \left[ f(G(Z)) \right]. \tag{3}$$

The term $\|X - G(Q(X))\|$ can be treated as the autoencoder reconstruction error as well as a loss to match the distributions between $X$ and $G(Q(X))$. We note that the $L_1$-norm $\|\cdot\|_1$ has been used for the reconstruction term by $\alpha$-GAN (Rosca et al., 2017) and CycleGAN (Zhu et al., 2017). Another term $\mathbb{E}_{X \sim P_X} f(G(Q(X))) - \mathbb{E}_{Z \sim P_Z} f(G(Z))$ can be treated as a loss for the generator as well as a loss to match the distribution between $G(Q(X))$ and $G(Z)$. We emphasize that this term is different with the objective function of the WGAN in (2). Furthermore, it is challenging for practitioners to determine when to stop training GANs. Most of the GAN algorithms do not provide any explicit standard for the convergence of the model. However, the measure of convergence for iWGAN becomes very natural and we use the duality gap as the measure. The duality gap can be defined as

$$\text{DualGap}(\widetilde{G}, \widetilde{Q}, \widetilde{f}) = \sup_{f \in \mathcal{F}} L(\widetilde{G}, \widetilde{Q}, f) - \inf_{G \in \mathcal{G}, Q \in \mathcal{Q}} L(G, Q, \widetilde{f}), \tag{4}$$

where $L(G, Q, f) = \mathbb{E}_X \|X - G(Q(X))\| + \mathbb{E}_X[f(G(Q(X)))] - \mathbb{E}_Z[f(G(Z))]$.

**Theorem 2.2.** *The iWGAN objective (3) is equivalent to*

$$\overline{W}_1(P_X, P_{G(Z)}) = \inf_{Q \in \mathcal{Q}} \left\{ W_1(P_X, P_{G(Q(X))}) + W_1(P_{G(Q(X))}, P_{G(Z)}) \right\}. \tag{5}$$

*Therefore, $W_1(P_X, P_{G(Z)}) \leq \overline{W}_1(P_X, P_{G(Z)})$. If there exists a $Q^* \in \mathcal{Q}$ such that $Q^*(X)$ has the same distribution with $Z$, then $W_1(P_X, P_{G(Z)}) = \overline{W}_1(P_X, P_{G(Z)})$. Let $(\widetilde{Q}, \widetilde{G}, \widetilde{f})$ be a fixed solution and assume that the encoder, generator, and discriminator all have enough capacities. Then the duality gap is larger than $W_1(P_X, P_{\widetilde{G}(\widetilde{Q}(X))}) + W_1(P_{\widetilde{G}(\widetilde{Q}(X))}, P_{\widetilde{G}(Z)})$. Moreover, if $\widetilde{G}$ outputs the same distribution as $X$ and $\widetilde{Q}$ outputs the same distribution as $Z$, both the duality gap and $\overline{W}_1(P_X, P_{\widetilde{G}(Z)})$ are zeros and $X = \widetilde{G}(\widetilde{Q}(X))$ for $X \sim P_X$.*

According to Theorem 2.2, the iWGAN objective is in general the upper bound of $W_1(P_X, P_{G(Z)})$. However, this upper bound is tight. When the space $\mathcal{Q}$ includes a special encoder $Q^*$ such that $Q^*(X)$ has the same distribution as $Z$, the iWGAN objective is the exactly same as $W_1(P_X, P_{G(Z)})$. Theorem 2.2 also provides an appealing property from a practical point of view. The values of the duality gap and $\overline{W}_1(P_X, P_{\widetilde{G}(Z)})$ give us a natural criteria to justify the algorithm convergence.

## 3 GENERALIZATION ERROR BOUND AND THE ALGORITHM

In practice, we minimize the empirical version, $\widehat{\overline{W}}_1(P_X, P_{G(Z)})$, of $\overline{W}_1(P_X, P_{G(Z)})$ to learn both the encoder and the decoder. Before we present the details of the algorithm, we first develop the generalization error bound for iWGANs. For discussions of generalization performance of classical GANs, see Arora et al. (2017) and Jiang et al. (2018).

**Theorem 3.1.** *Given a generator $G \in \mathcal{G}$, and given $n$ samples $(x_1, \ldots, x_n)$ from $\mathcal{X} = \{x : \|x\| \leq B\}$, with probability at least $1 - \delta$ for any $\delta \in (0, 1)$, we have*

$$W_1(P_X, P_{G(Z)}) \leq \widehat{\overline{W}}_1(P_X, P_{G(Z)}) + 2\widehat{\mathfrak{R}}_n(\mathcal{F}) + 3B\sqrt{\frac{2}{n} \log\left(\frac{2}{\delta}\right)}, \tag{6}$$

*where $\widehat{\mathfrak{R}}_n(\mathcal{F}) = \mathbb{E}_\epsilon \left[ \sup_{f \in \mathcal{F}} n^{-1} \sum_{i=1}^n \epsilon_i f(x_i) \right]$ is the empirical Rademacher complexity of the 1-Lipschitz function set $\mathcal{F}$, in which $\epsilon_i$ is the Rademacher variable.*

Theorem 3.1 indicates that the 1-Wasserstein distance between $P_X$ and $P_{G(Z)}$ can be dominantly upper bounded by the empirical $\widehat{\overline{W}}_1(P_X, P_{G(Z)})$ and Rademacher complexity of $\mathcal{F}$. Since $\widehat{\overline{W}}_1(P_X, P_{G(Z)}) \leq \widehat{W}_1(P_X, P_{G(Q(X))}) + \widehat{W}_1(P_{G(Q(X))}, P_{G(Z)})$ for any $Q \in \mathcal{Q}$, the capacity of $\mathcal{Q}$ determines the value of $\widehat{\overline{W}}_1(P_X, P_{G(Z)})$. On the other hand, there are several existing results on the empirical Rademacher complexity of neural networks. When $\mathcal{F}$ is a set of 1-Lipschitz neural networks, we can apply the conclusion from Bartlett et al. (2017) to $\widehat{\mathfrak{R}}_n(\mathcal{F})$, which produces an upper bound scaling as $\mathcal{O}(B\sqrt{L^3/n})$. Here $L$ denotes the depth of network $f \in \mathcal{F}$. Similar upper bound with an order of $\mathcal{O}(B\sqrt{Ld^2/n})$ can be obtained by utilizing the results from Li et al. (2018), where $d$ is the width of the network.

In practice, we adopt the gradient penalty defined as $\mathcal{GP}(f) = E_X\left[(\|\nabla_X f(X)\|_2 - 1)^2\right]$ in (Gulrajani et al., 2017) to enforce the 1-Lipschitz constraint on $f \in \mathcal{F}$. In addition, we use the maximum mean discrepancy (MMD) penalty (Gretton et al., 2012), denoted by $\text{MMD}_k(P_{Q(X)}, P_Z)$, to enforce $Q(X)$ to converge to $P_Z$, where $k$ is a kernel function. The details of the algorithm are presented in Algorithm 1.

---

**Algorithm 1:** The training algorithm of iWGAN

1 **while** *DualGap* $> \epsilon_1$ *or* $L(G^i, Q^i, f^i) > \epsilon_2$ **do**
2    **for** $t = 1, ..., n_{critic}$ **do**
3       Sample real data $x^i \sim P_X$, latent variable $z^i \sim P_Z$ and a random number $\epsilon \sim U[0,1]$
4       $\hat{x}^i \leftarrow \epsilon x^i + (1-\epsilon)G^i(z^i)$
5       Calculate $L^i = L(G^i, Q^i, f^i | x^i, z^i)$ and gradient of $-L^i$
6       Update $f$ by Adam: $f^{i+1} \leftarrow Adam(-\nabla_f L^i)$
7       where for $f^i$,
8       $-\nabla_f L^i = \nabla_f \frac{1}{n}\sum_{k=1}^n \left( f^i(G^i(z_k^i)) - f^i(G^i(Q^i(x_k^i))) + \lambda_1(\|\nabla_{\hat{x}^i} f^i(\hat{x}^i)\|_2 - 1)^2 \right)$
9    **end**
10   **for** $t = 1, ..., n_{critic}$ **do**
11      Sample real data $x'^i \sim P_X$, latent variable $z'^i \sim P_Z$
12      Calculate $L'^i = L(G^i, Q^i, f^{i+1} | x'^i, z'^i)$ and gradient of $L'^i$
13      Update $G$, $Q$ by Adam: $G^{i+1}, Q^{i+1} \leftarrow Adam(\nabla_{G,Q} L'^i)$
14      where for $G^i, Q^i$,

$$\nabla_{G,Q} L'^i = \nabla_{G,Q} \frac{1}{n}\sum_{k=1}^n (\|x_k^i - G^i(Q^i(x_k^i))\| + f^{i+1}(G^i(Q^i(x_k^i))) - f^{i+1}(G^i(z_k^i)))$$
$$+ \frac{\lambda_2}{n(n-1)}\sum_{l \neq j} k(z_l^i, z_j^i) + \frac{\lambda_2}{n(n-1)}\sum_{l \neq j} k(Q(x_l^i), Q(x_j^i)) - \frac{2\lambda_2}{n^2}\sum_{l,j} k(z_l^i, Q(x_j^i))$$

15   **end**
16 **end**

---

## 4 PROBABILISTIC INTERPRETATION AND THE MLE

The iWGAN has proposed an efficient framework to stably and automatically estimate both the encoder and the generator. In this section, we provide a probabilistic interpretation of the iWGAN under the framework of maximum likelihood estimation.

Maximum likelihood estimator (MLE) is a fundamental statistical framework for learning models from data. However, for complex models, MLE can be computationally prohibitive due to the intractable normalization constant. MCMC has been used to approximate the intractable likelihood function but do not work efficiently in practice. The iWGAN can be treated as an adaptive method for MLE training, which not only provides computational advantages but also allows us to generate more realistic-looking images. Furthermore, this probabilistic interpretation enables other novel applications such as image quality checking and outlier detection.

Let $X$ denote the image. Define the density of $X$ by an energy-based model based on an autoencoder (Zhao et al., 2016; Berthelot et al., 2017):

$$p(x|\theta) = \exp\left(-\|x - G_\theta(Q_\theta(x))\| - V(\theta)\right), \quad V(\theta) = \log \int \exp(-\|x - G_\theta(Q_\theta(x))\|)dx,$$

where $\theta$ is the unknown parameter and $V(\theta)$ is the log normalization constant. The major difficulty for the likelihood inference is due to the intractable function $V(\theta)$. Suppose that we have the observed data $\{x_i : i = 1, \ldots, n\}$. The log-likelihood function of $\theta$ is $\ell(\theta) = n^{-1}\sum_{i=1}^n \log p(x_i|\theta)$, whose gradient is

$$\nabla_\theta \ell(\theta) = -\hat{\mathbb{E}}_{obs}\left[\partial_\theta \|x - G_\theta(Q_\theta(x))\|\right] + \mathbb{E}_\theta\left[\partial_\theta \|x - G_\theta(Q_\theta(x))\|\right], \tag{7}$$

where $\hat{\mathbb{E}}_{obs}[\cdot]$ denotes the empirical average on the observed data $\{x_i\}$ and $\mathbb{E}_\theta[\cdot]$ denotes the expectation under model $p(x|\theta)$. The key computational obstacle lies in the approximations of the

model expectation $\mathbb{E}_\theta[\cdot]$. To address this problem, we propose a novel dual approximation for this expectation. By Theorem 5.10 of Villani (2008), there exists an optimal $f^*$ such that

$$\mathbb{P}_{(x,y)\sim\pi}\left[f^*(y) - f^*(x) = \|y - x\|\right] = 1 \tag{8}$$

for the optimal coupling $\pi$. Therefore, there exists a $f^*$ such that $f^*(x) - f^*(G_\theta(Q_\theta(x))) = \|x - G_\theta(Q_\theta(x))\|$ with probability one with respect to the distribution of $x$. Since $\mathbb{E}_\theta$ in (7) is taken under the current estimated $\theta$ and we also require $G_\theta$ to be a good generator and the distributions of $G_\theta(z)$ and $G_\theta(Q_\theta(x))$ to be close, we approximate $\|x - G_\theta(Q_\theta(x))\|$ by $f^*(G_\theta(z)) - f^*(G_\theta(Q_\theta(x)))$. We replace $\|x - G_\theta(Q_\theta(x))\|$ in the second term of (7) by $f^*(G_\theta(z)) - f^*(G_\theta(Q_\theta(x)))$, yielding a gradient update for $\theta$ of form $\theta \leftarrow \theta + \epsilon\hat{\nabla}_\theta\ell(\theta)$, where

$$\hat{\nabla}_\theta\ell(\theta) = -\hat{\mathbb{E}}_{obs}\left[\partial_\theta\|x - G_\theta(Q_\theta(x))\|\right] + \mathbb{E}_\theta\left[\partial_\theta f^*(G_\theta(z)) - \partial_\theta f^*(G_\theta(Q_\theta(x)))\right]. \tag{9}$$

Here $f^*$ needs to be learned and is solved by the corresponding dual problem at each iteration. We approximate $f^*$ by a network $f_\eta$ with an unknown parameter $\eta$, yielding a gradient update for $\eta$ of form

$$\eta \leftarrow \eta + \epsilon\,\mathbb{E}_\theta\left[\partial_\eta f_\eta(G_\theta(z)) - \partial_\eta f_\eta(G_\theta(Q_\theta(x)))\right]. \tag{10}$$

The advantage of using expectations in (9) and (10) is that we can evaluate them by using only marginal distributions of $z$ and $x$. The above iterative updating process is exactly the same as in Algorithm 1. Therefore, the training of iWGAN is to seek the MLE. This probabilistic interpretation provides a novel alternative method to tackle problems with the intractable normalization constant in latent variable models. The MLE gradient update of $p(x|\theta)$ decreases the energy of the training data and increases the dual objective. Compare with original GANs or WGANs, our method gives much faster convergence and simultaneously provides a higher quality generated images.

The probabilistic modeling opens a door for many interesting applications. Next, we present a completely new approach for determining a highest density region (HDR) estimate for the distribution of $X$. What makes HDR distinct from other statistical methods is that it finds the smallest region, denoted by $U(\alpha)$, in the high dimensional space with a given probability coverage $1 - \alpha$, i.e., $\mathbb{P}(X \in U(\alpha)) = 1 - \alpha$. We can use $U(\alpha)$ to assess each individual sample quality. Note that FID or the Inception score are used to measure the whole sample quality, not at the individual sample level. Let $\hat{\theta}$ be the MLE. The density ratio at $x_1$ and $x_2$ is

$$\frac{p(x_1|\hat{\theta})}{p(x_2|\hat{\theta})} = \exp\left[-\left(\|x_1 - G_{\hat{\theta}}(Q_{\hat{\theta}}(x_1))\| - \|x_2 - G_{\hat{\theta}}(Q_{\hat{\theta}}(x_2))\|\right)\right].$$

The smaller the reconstruction error is, the larger the density value is. We can define the HDR for $x$ through the HDR for the reconstruction error $e_x := \|x - G_{\hat{\theta}}(Q_{\hat{\theta}}(x))\|$, which is simple because it is a one-dimensional problem. Let $\tilde{U}(\alpha)$ be the HDR for $e_x$. Then, $U(\alpha) = \{x : e_x \in \tilde{U}(\alpha)\}$. Here $Q_{\hat{\theta}}(U(\alpha))$ defines the corresponding region in the latent space, which can be used to generate better quality samples.

# 5 EXPERIMENTAL RESULTS

## 5.1 MIXTURE OF GAUSSIANS

We train our iWGAN model on three toy datasets with an increasing difficulty shown on the right:

a). RING: a mixture of 8 Gaussians, b). SPIRAL: a mixture of 20 Gaussians and c). GRID: a mixture of 25 Gaussians. As the true data distributions are known, this setting allows for tracking of convergence and mode dropping.

**Duality gap and convergence:** We illustrate that as the duality gap converges to 0, our model converges to

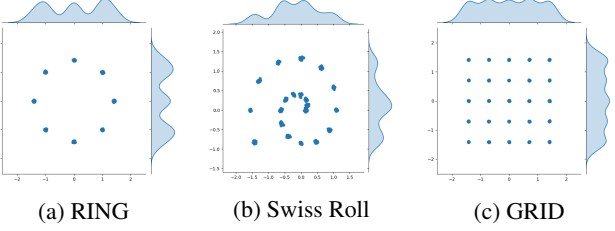

(a) RING      (b) Swiss Roll      (c) GRID

the generated samples from the true distribution. We keep track of the generated samples using $G(z)$ and record the duality gap at each iteration to check the corresponding generated samples. Figure 2 shows the generated samples converge to the true distribution very fast without the mode collapse

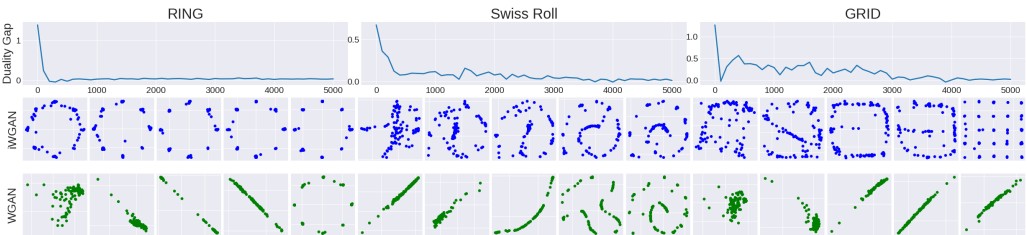

Figure 2: Duality gap and generated samples from iWGANs on mixture of Gaussians

problem. We compare our method with WGAN-GP in Figure 2. Both methods adopt the same structure, learning rate, number of critical steps, and other hyper-parameters. The iWGAN surpasses the performance of the WGAN-GP at very early stage and avoids the appearance of mode collapse.

**Latent Space:** We choose the latent distribution to be a five dimensional standard normal distribution $Z \sim N(0, I_5)$. After training, the distribution of $Q(X)$ is expected to be close to the distribution of $Z$. We plot the $Q(X)_i$ against $Q(X)_j$ for all $i \neq j$ in Figure 3. We can tell that the joint distribution of any two dimensions of $Q(X)$ is close to a bivariate normal distribution.

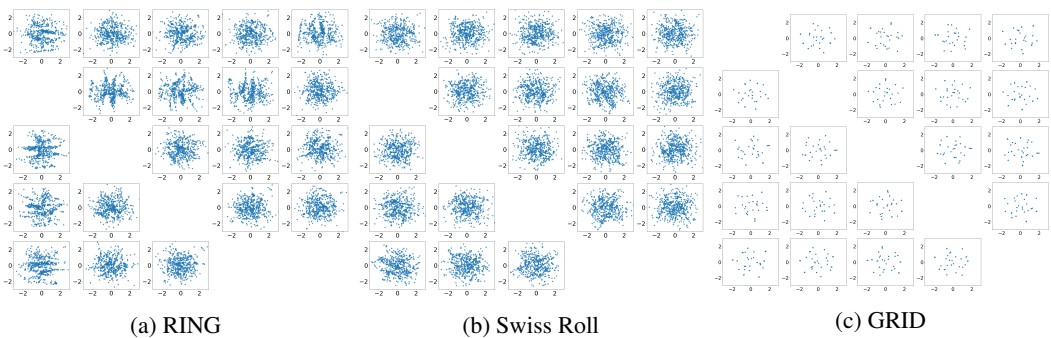

(a) RING        (b) Swiss Roll        (c) GRID

Figure 3: Latent Space of Mixture of Gaussians

**Individual sample quality check:** From the probability interpretation of iW-GANs, we naturally adopt the reconstruction error $\|X - G(Q(X))\|$, or the *quality score* $\exp\left(-\|X - G(Q(X))\|\right)$ as the metric of the quality of any individual sample. The larger the quality score is, the better quality the sample has. Figure 4 shows their quality scores for different samples. The quality scores of samples near the modes of the true distribution are close to 1, and become smaller as the sample draw is away from the modes. This indicates that the iWGAN does converge and learns the distribution well, and the *quality score* is a reliable metric for the individual sample quality.

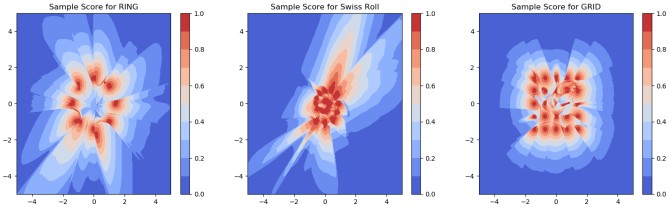

Figure 4: Quality Check

### 5.2 MNIST & CELEBA

We experimentally demonstrate our model's ability on well-known datasets, MNIST and CelebA. The results by iWGAN on MNIST and CelebA are shown in Figure 5 and Figure 6, respectively. For latent space interpolations between MNIST or CelebA validation set examples. We sample pairs of validation set examples $x_1$ and $x_2$ and project them into $z_1$ and $z_2$ by the encoder. We then linearly interpolate between $z_1$ and $z_2$ and pass the intermediary points through the decoder to plot the input-space interpolations. Figure 7

displays images with high and low quality scores selected from CelebA. More experimental results, including reconstructed images, interpolations, and comparison with state-of-the-art results by WGAN-GP, are shown in Appendix.

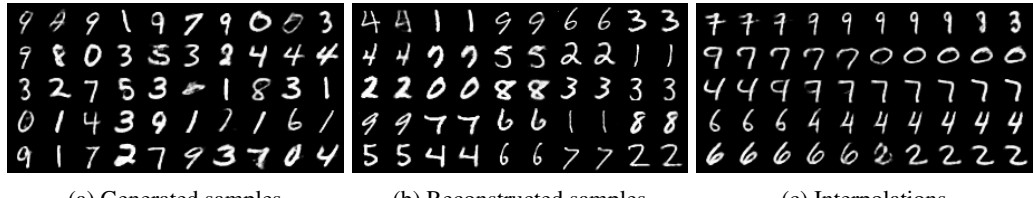

(a) Generated samples      (b) Reconstructed samples      (c) Interpolations

Figure 5: iWGAN on MNIST

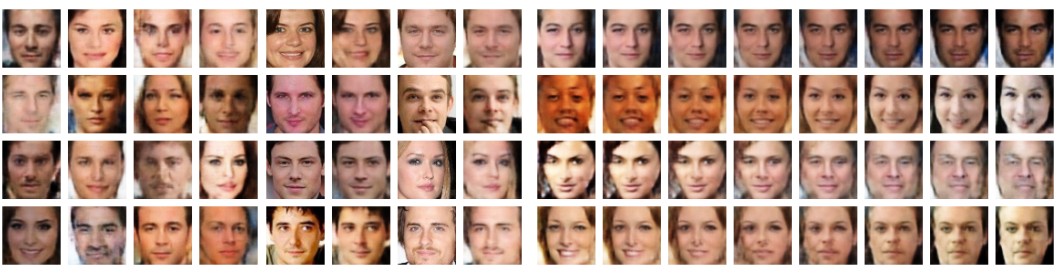

(a) Generated samples      (b) Reconstructed samples      (c) Interpolations

Figure 6: iWGAN on CelebA

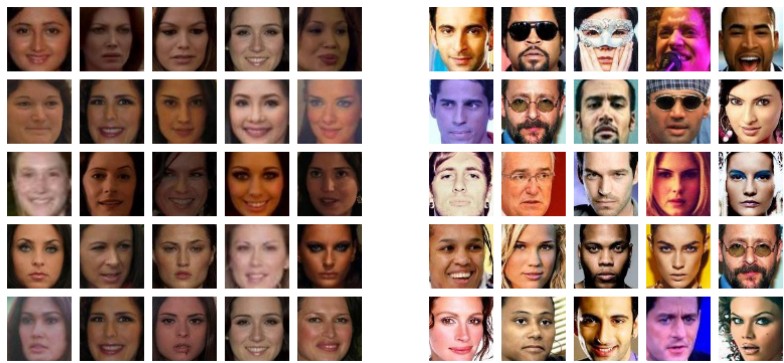

Figure 7: Images with high (left) and low (right) quality scores by iWGAN

## 6 CONCLUSION

We have developed a novel iWGAN model, which fuses auto-encoders and GANs in a principle way. We have established a generalization error bound for iWGAN. We have provided a solid probabilistic interpretation on iWGAN using the maximum likelihood principle. Our training algorithm with an iterative primal and dual optimization has demonstrated an efficient and stable learning. We have proposed a stopping criteria for our algorithm and a metric for individual sample quality checking. The empirical results on both synthetic and benchmark datasets are state-of-the-art.

We now mention several future directions for research on iWGAN. First, one might be interested in applying iWGAN into image-to-image translation, as the extension should be straightforward. A second direction is to develop a formal hypothesis testing procedure to test whether the samples generated from iWAGN is the same as the data distribution.

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

APPENDIX

## APPENDIX A: PROOF OF THEOREM 2.1

According to the Nash embedding theorem (Nash, 1956; Günther, 1991), every $d$-dimensional smooth Riemannian manifold $\mathcal{X}$ possesses a smooth isometric embedding into $\mathbb{R}^p$ with $p = \max\{d(d+5)/2, d(d+3)/2+5\}$. Therefore, there exists an injective mapping $u : \mathcal{X} \to \mathbb{R}^p$ which preserves the metric in the sense that the manifold metric on $\mathcal{X}$ is equal to the pullback of the usual Euclidean metric on $\mathbb{R}^p$ by $u$. The mapping $u$ is injective so that we can define the inverse mapping $u^{-1} : u(\mathcal{X}) \to \mathcal{X}$.

Let $\tilde{X} = u(X) \in \mathbb{R}^p$, and write $\tilde{X} = (\tilde{X}_1, \ldots, \tilde{X}_p)$. Let $F_i(x) = \mathbb{P}(\tilde{X}_i \leq x)$, $i = 1, \ldots, p$, be the marginal cdfs. By applying the probability integral transformation to each component, the random vector

$$\big(U_1, U_2, \ldots, U_p\big) := \big(F_1(\tilde{X}_1), F_2(\tilde{X}_2), \ldots, F_p(\tilde{X}_p)\big)$$

has uniformly distributed marginals. Let $C : [0,1]^p \to [0,1]$ be the copula of $\tilde{X}$, which is defined as the joint cdf of $(U_1, \ldots, U_p)$:

$$C(u_1, u_2, \ldots, u_p) = \mathbb{P}\big(U_1 \leq u_1, U_2 \leq u_2, \ldots, U_p \leq u_p\big).$$

The copula $C$ contains all information on the dependence structure among the components of $\tilde{X}$, while the marginal cumulative distribution functions $F_i$ contain all information on the marginal distributions. Therefore, the joint cdf of $\tilde{X}$ is

$$H(\tilde{x}_1, \tilde{x}_2, \ldots, \tilde{x}_p) = C\big(F_1(\tilde{x}_1), F_2(\tilde{x}_2), \ldots, F_p(\tilde{x}_p)\big).$$

Define, for $i = 2, \ldots, p$,

$$C_i(u_1, u_2, \ldots, u_i) = C\big(u_1, u_2, \ldots, u_i, 1, \ldots, 1\big).$$

The conditional distribution of $U_k$, given $U_1, \ldots, U_{k-1}$, is given by Cherubini et al. (2004)

$$\begin{aligned}
C_k(u_k | u_1, \ldots, u_{k-1}) &= \mathbb{P}\big(U_k \leq u_k | U_1 = u_1, \ldots, U_{k-1} = u_{k-1}\big) \\
&= \frac{\big[\partial^{k-1} C_k(u_1, \ldots, u_k)/\partial u_1 \cdots \partial u_{k-1}\big]}{\big[\partial^{k-1} C_{k-1}(u_1, \ldots, u_k)/\partial u_1 \cdots \partial u_{k-1}\big]},
\end{aligned}$$

for $k = 2, \ldots, p$.

We will construct $Q^*$ as follows. First, we obtain $\tilde{X} \in \mathbb{R}^p$ by $\tilde{X} = u(X)$. Second, we transform $\tilde{X}$ into a random vector with uniformly distributed marginals $(U_1, \ldots, U_p)$ by the marginal cdf $F_i$. Then, define $\tilde{U}_1 = U_1$ and

$$\tilde{U}_k = C_k\big(U_k | U_1, \ldots, U_{k-1}\big), \quad k = 2, \ldots, p.$$

Hence, $\tilde{U}_1, \ldots, \tilde{U}_p$ are independent uniform random variables. Finally, let $Z_i = \Phi^{-1}(U_i)$ for $i = 1, \ldots, p$. This completes the transformation $Q^*$ from $X$ to $Z = (Z_1, \ldots, Z_p)$.

The above process can be inverted to obtain $G^*$. First, we transform $Z$ into independent uniform random variables by $\tilde{U}_i = \Phi(Z_i)$ for $i = 1, \ldots, p$. Next, let $U_1 = \tilde{U}_1$. Define

$$U_k = C_k^{-1}(\tilde{U}_k | \tilde{U}_1, \ldots, \tilde{U}_{k-1}), \quad i = 2, \ldots, p,$$

where $C_k^{-1}(\cdot | u_1, \ldots, u_k)$ is the inverse of $C_k$ and can be obtained by numerical root finding. Finally, let $\tilde{X}_i = F_i^{-1}(U_i)$ for $i = 1, \ldots, p$ and $X = u^{-1}(\tilde{X})$, where $u^{-1} : u(\mathcal{X}) \to \mathcal{X}$ is the inverse mapping of $u$. This completes the transformation $G^*$ from $Z$ to $X$.

## APPENDIX B: PROOF OF THEOREM 2.2

By the iWGAN objective (3), (5) holds. Since $W_1$ is a distance between two probability measures, $W_1(P_X, P_{G(Z)}) \leq \overline{W}_1(P_X, P_{G(Z)})$. If there exists a $Q^* \in \mathcal{Q}$ such that $Q^*(X)$ has the same distribution as $P_Z$, we have

$$\overline{W}_1(P_X, P_{G(Z)}) \leq W_1(P_X, P_{G(Q^*(X))}) + W_1(P_{G(Q^*(X))}, P_{G(Z)}) = W_1(P_X, P_{G(Z)}).$$

Hence, $W_1(P_X, P_{G(Z)}) = \overline{W}_1(P_X, P_{G(Z)})$. Observe that $\sup_f L(\widetilde{G}, \widetilde{Q}, f) = W_1(P_X, P_{\widetilde{G}(\widetilde{Q}(X))}) + W_1(P_{\widetilde{G}(\widetilde{Q}(X))}, P_{\widetilde{G}(Z)})$. By Theorem 2.1, we have $\inf_{G,Q} L(G, Q, \widetilde{f}) \leq L(G^*, Q^*, \widetilde{f}) = 0$ when $\mathcal{G}$ and $\mathcal{Q}$ have enough capacity. Therefore, the duality gap is larger than $W_1(P_X, P_{\widetilde{G}(\widetilde{Q}(X))}) + W_1(P_{\widetilde{G}(\widetilde{Q}(X))}, P_{\widetilde{G}(Z)})$. It is easy to see that, if $\widetilde{G}$ outputs the same distribution as $X$ and $\widetilde{Q}$ outputs the same distribution as $Z$, both the duality gap and $\overline{W}_1(P_X, P_{G(Z)})$ are zeros and $X = \widetilde{G}(\widetilde{Q}(X))$ for $X \sim P_X$.

## APPENDIX C: PROOF OF THEOREM 3.1

We first consider the difference between population $W_1(P_X, P_{G(Z)})$ and empirical $\widehat{W}_1(P_X, P_{G(Z)})$ given $n$ samples $S = \{x_1, \ldots, x_n\}$. Let $f_1$ and $f_2$ be their witness function respectively. Using the dual form of 1-Wassertein distance, we have

$$W_1(P_X, P_{G(Z)}) - \widehat{W}_1(P_X, P_{G(Z)})$$

$$= \mathbb{E}_{X \sim P_X}[f_1(X)] - \mathbb{E}_{Z \sim P_Z}[f_1(G(Z))] - \frac{1}{n}\sum_{i=1}^{n} f_2(x_i) + \mathbb{E}_{Z \sim P_Z}[f_2(G(Z))]$$

$$\leq \mathbb{E}_{X \sim P_X}[f_1(X)] - \mathbb{E}_{Z \sim P_Z}[f_1(G(Z))] - \frac{1}{n}\sum_{i=1}^{n} f_1(x_i) + \mathbb{E}_{Z \sim P_Z}[f_1(G(Z))]$$

$$\leq \sup_f \mathbb{E}_{X \sim P_X}[f(X)] - \frac{1}{n}\sum_{i=1}^{n} f(x_i) \triangleq \Phi(S).$$

Given another sample set $S' = \{x_1, \ldots, x_i', \ldots, x_n\}$, it is clear that

$$\Phi(S) - \Phi(S') \leq \sup_f \frac{|f(x_i) - f(x_i')|}{n} \leq \frac{\|x_i - x_i'\|}{n} \leq \frac{2B}{n},$$

where the second inequality is obtained since $f$ is 1-Lipschitz continuous function. Applying McDiamond's Inequality, with probability at least $1 - \delta/2$ for any $\delta \in (0, 1)$, we have

$$\Phi(S) \leq \mathbb{E}[\Phi(S)] + B\sqrt{\frac{2}{n}\log\left(\frac{2}{\delta}\right)}. \tag{11}$$

By the standard technique of symmetrization in Mohri et al. (2018), we have

$$\mathbb{E}[\Phi(S)] = \mathbb{E}\left[\sup_f \mathbb{E}_{X \sim P_X}[f(X)] - \frac{1}{n}\sum_{i=1}^{n} f(x_i)\right] \leq 2\mathfrak{R}_n(\mathcal{F}). \tag{12}$$

It has been proved in Mohri et al. (2018) that with probability at least $1 - \delta/2$ for any $\delta \in (0, 1)$,

$$\mathfrak{R}_n(\mathcal{F}) \leq \widehat{\mathfrak{R}}_n(\mathcal{F}) + B\sqrt{\frac{2}{n}\log\left(\frac{2}{\delta}\right)}. \tag{13}$$

Combining Equation (11), Equation (12) and Equation (13), we have

$$W_1(P_X, P_{G(Z)}) \leq \widehat{W}_1(P_X, P_{G(Z)}) + 2\widehat{\mathfrak{R}}_n(\mathcal{F}) + 3B\sqrt{\frac{2}{n}\log\left(\frac{2}{\delta}\right)}.$$

By Theorem 2.2, we have $\widehat{W}_1(P_X, P_{G(Z)}) \leq \widehat{\overline{W}}_1(P_X, P_{G(Z)})$. Thus,

$$W_1(P_X, P_{G(Z)}) \leq \widehat{\overline{W}}_1(P_X, P_{G(Z)}) + 2\widehat{\mathfrak{R}}_n(\mathcal{F}) + 3B\sqrt{\frac{2}{n}\log\left(\frac{2}{\delta}\right)}.$$

## APPENDIX D: EXTENSION TO $f$-GANS

This framework can be easily extended to other types of GANs. We consider $f$-GAN (Nowozin et al., 2016) here. Let $h : \mathbb{R} \to (-\infty, \infty]$ be a convex function with $h(1) = 0$. The $f$-GAN minimizes the following objective for the generator $G$:

$$\text{GAN}_h(P_X, P_{G(Z)}) = \sup_{f \in \mathcal{F}} \left\{\mathbb{E}_X[f(X)] - \mathbb{E}_Z[h^*(f(G(Z)))]\right\},$$

where $h^*(x) = \sup_y\{x \cdot y - h(y)\}$ is the convex conjugate of $h$. If $h(x) = x\log(x) - (x+1)\log(x+1) - 2\log 2$, $f$-GAN recovers the original GAN (Goodfellow et al., 2014). If $h(x) = 0$ when $x = 1$ and $h(x) = \infty$ otherwise, we have $h^*(x) = x$. With the property that $\mathcal{F}$ is 1-Lipschitz function class, $f$-GAN turns to be WGAN.

Assume that $\mathcal{F}$ is the 1-Lipschitz function class. We extent the iWGAN framework to the inference f-GAN (ifGAN) framework and define the new objective function as follows:

$$\overline{W}_{1,h}(P_X, P_{G(Z)}) = \inf_{Q \in \mathcal{Q}} \sup_{f \in \mathcal{F}} \mathbb{E}_X\|X - G(Q(X))\| + \mathbb{E}_X[f(G(Q(X)))] - \mathbb{E}_Z[h^*(f(G(Z)))]. \tag{14}$$

Following this definition, we have
$$\overline{W}_{1,h}(P_X, P_{G(Z)}) = \inf_{Q \in \mathcal{Q}} \left\{ W_1(P_X, P_{G(Q(X))}) + \text{GAN}_h(P_{G(Q(X))}, P_{G(Z)}) \right\}.$$

We show $\text{GAN}_h(P_X, P_{G(Z)}) \leq \overline{W}_{1,h}(P_X, P_{G(Z)})$. This is because
$$\text{GAN}_h(P_X, P_{G(Z)}) = \sup_{f \in \mathcal{F}} \mathbb{E}_X\left[f(X)\right] - \mathbb{E}_Z\left[h^*(f(G(Z)))\right]$$

$$\leq \inf_{Q \in \mathcal{Q}} \left\{ \sup_{f \in \mathcal{F}} \mathbb{E}_X\left[f(X)\right] - \mathbb{E}_X\left[f(G(Q(X)))\right] + \sup_{f \in \mathcal{F}} \mathbb{E}_X\left[f(G(Q(X)))\right] - \mathbb{E}_Z\left[h^*(f(G(Z)))\right] \right\}$$

$$= \overline{W}_{1,h}(P_X, P_{G(Z)}).$$

This indicates that the ifGAN objective (14) is an upper bound of the f-GAN objective.

## APPENDIX E: ARCHITECTURES [1]

### E1: MIXTURE OF GUASSIANS

Encoder architecture:
$$x \in \mathbb{R}^2 \rightarrow FC_{1024} \rightarrow RELU$$
$$\rightarrow FC_{512} \rightarrow RELU$$
$$\rightarrow FC_{256} \rightarrow RELU$$
$$\rightarrow FC_{128} \rightarrow RELU \rightarrow FC_5$$

Generator architecture:
$$z \in \mathbb{R}^5 \rightarrow FC_{512} \rightarrow RELU$$
$$\rightarrow FC_{512} \rightarrow RELU$$
$$\rightarrow FC_{512} \rightarrow RELU \rightarrow FC_2$$

Discriminator architecture:
$$x \in \mathbb{R}^2 \rightarrow FC_{512} \rightarrow RELU$$
$$\rightarrow FC_{512} \rightarrow RELU$$
$$\rightarrow FC_{512} \rightarrow RELU \rightarrow FC_1$$

### E2: MNIST

Encoder architecture:
$$x \in \mathbb{R}^{28 \times 28} \rightarrow Conv_{128} \rightarrow RELU$$
$$\rightarrow Conv_{256} \rightarrow RELU$$
$$\rightarrow Conv_{512} \rightarrow RELU \rightarrow FC_8$$

Generator architecture:
$$z \in \mathbb{R}^8 \rightarrow FC_{4 \times 4 \times 512} \rightarrow RELU$$
$$\rightarrow ConvTrans_{256} \rightarrow RELU$$
$$\rightarrow ConvTrans_{128} \rightarrow RELU \rightarrow ConvTrans_1$$

Discriminator architecture:
$$x \in \mathbb{R}^{28 \times 28} \rightarrow Conv_{128} \rightarrow RELU$$
$$\rightarrow Conv_{256} \rightarrow RELU$$
$$\rightarrow Conv_{512} \rightarrow RELU \rightarrow FC_1$$

---

[1]Codes used for this paper will be available at: `https://drive.google.com/drive/folders/1-_vIrbOYwf2BH1lOrVEcEPJUxkyV5CiB?usp=sharing`

E3: CELEBA

Encoder architecture:

$$x \in \mathbb{R}^{64 \times 64 \times 3} \to Conv_{128} \to LeakyRELU$$
$$\to Conv_{256} \to InstanceNorm \to LeakyRELU$$
$$\to Conv_{512} \to InstanceNorm \to LeakyRELU \to Conv_1$$

Generator architecture:

$$z \in \mathbb{R}^{64} \to FC_{4 \times 4 \times 1024}$$
$$\to ConvTrans_{512} \to BN \to RELU$$
$$\to ConvTrans_{256} \to BN \to RELU$$
$$\to ConvTrans_{128} \to BN \to RELU \to ConvTrans_3$$

Discriminator architecture:

$$x \in \mathbb{R}^{64 \times 64 \times 3} \to Conv_{128} \to LeakyRELU$$
$$\to Conv_{256} \to InstanceNorm \to LeakyRELU$$
$$\to Conv_{512} \to InstanceNorm \to LeakyRELU \to Conv_1$$

## APPENDIX F: MORE EXPERIMENTAL RESULTS ON MIXTURE OF GAUSSIAN

We investigate the mode collapse problem for the iWGAN. If we draw two random samples in the latent space $z_1, z_2 \sim N(0, I_5)$, the interpolation, $G(\lambda z_1 + (1 - \lambda)z_2), 0 \leq \lambda \leq 1$, should fall around the mode to represent a reasonable sample. In Figure 8, we select $\lambda \in \{0, 0.05, 0.10, \dots, 0.95, 1.0\}$, and do interpolations on two random samples. We repeat this procedure several times on 3 datasets as demonstrated in Figure 8. No matter where the interpolations start and end, the interpolations would fall around the modes other than the locations where true distribution has a low density. There may still be some samples that appears in the middle of two modes. This may be because the generator $G$ is not able to approximate a step function well.

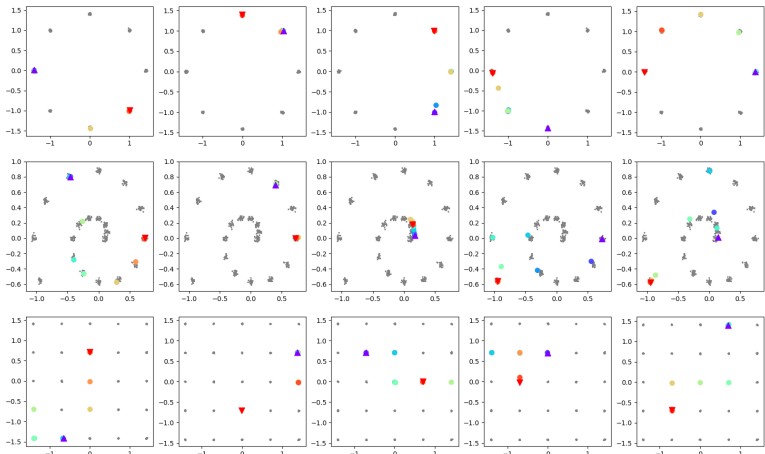

Figure 8: Interpolation: ▼ and ▲ indicates the first and last samples in the interpolations, other colored samples are the interpolations.

## APPENDIX G: MORE EXPERIMENTAL RESULTS ON MNIST

### G.1: LATENT SPACE

Figure 9 shows the latent space of MNIST, i.e. $Q(X)_i$ against $Q(X)_j$ for all $i \neq j$.

### G.2: GENERATED SAMPLES

Figure 10 shows the comparison of random generated samples between WGAN-GP and iWGAN. Figure 11 shows examples of interpolations of two random generated samples.

### G.3: RECONSTRUCTION

Figure 12 shows, based on the samples from validation dataset, the distribution of reconstruction error. Figure 13 shows examples of reconstructed samples. Figure 14 shows the best and worst samples based on quality scores from the validation dataset.

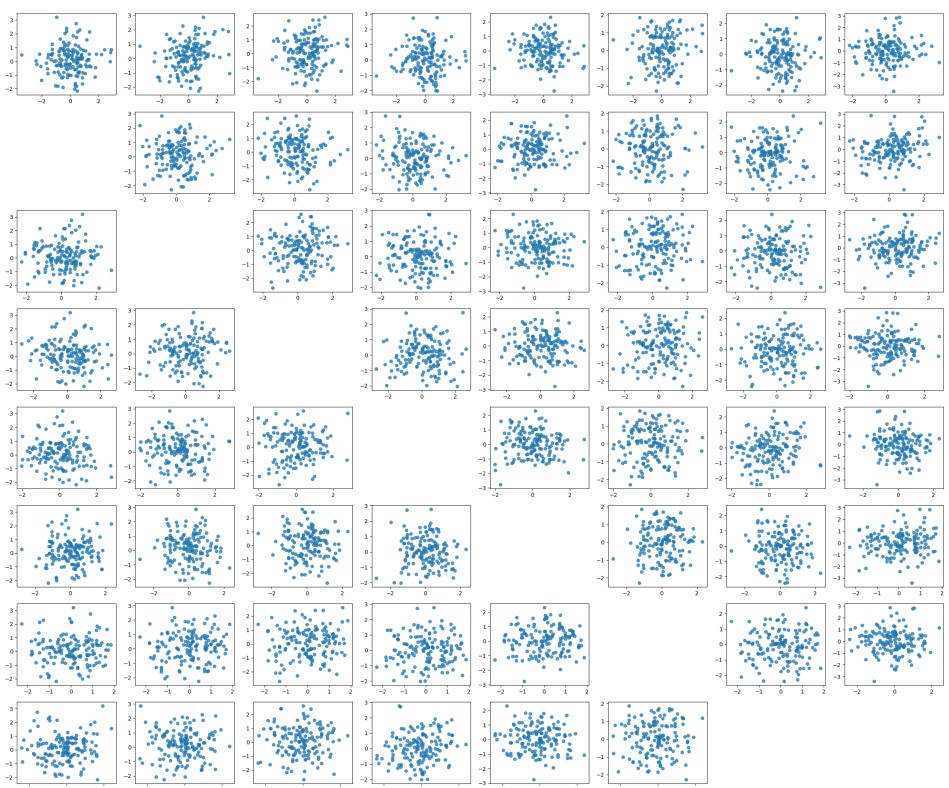

Figure 9: Latent Space of MNIST dataset

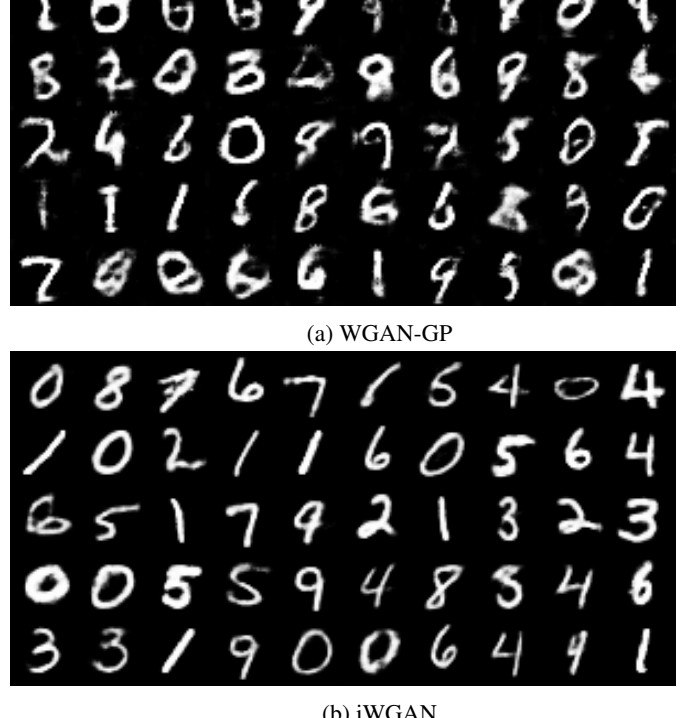

(a) WGAN-GP

(b) iWGAN

Figure 10: Generated samples on MNIST

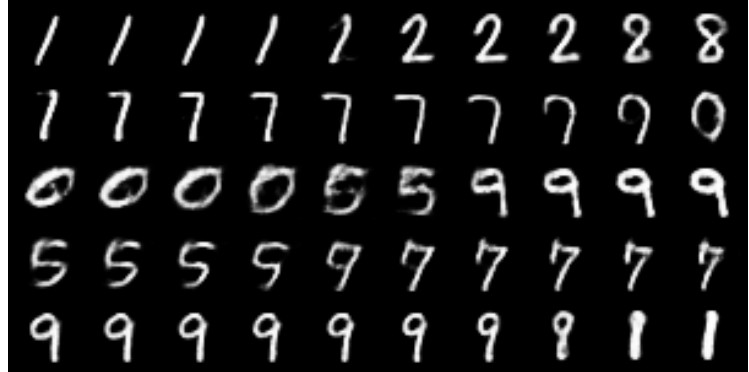

Figure 11: Interpolations on MNIST

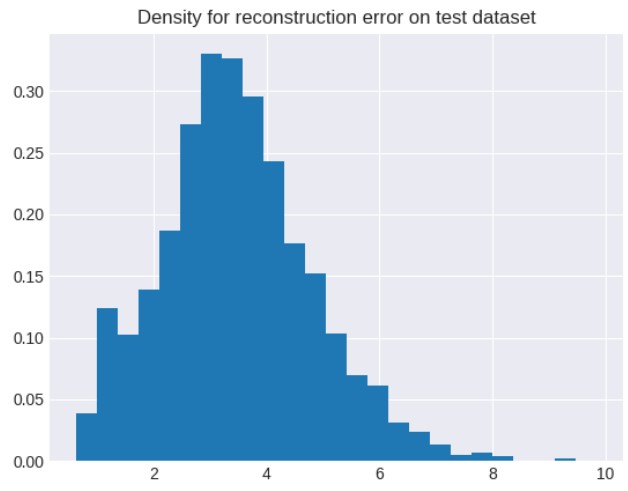

Figure 12: Histogram of reconstruction error on MNIST

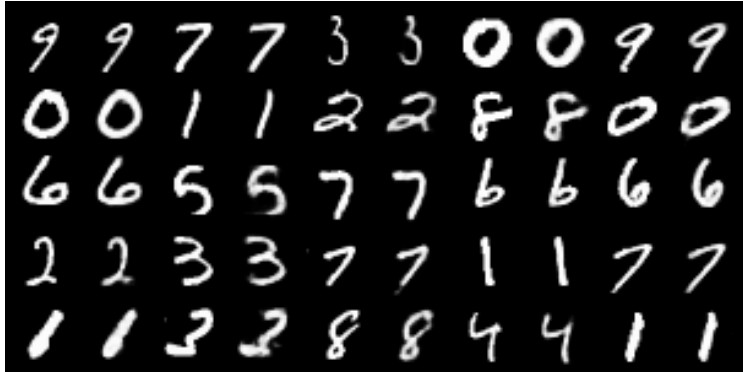

Figure 13: Reconstructions on MNIST

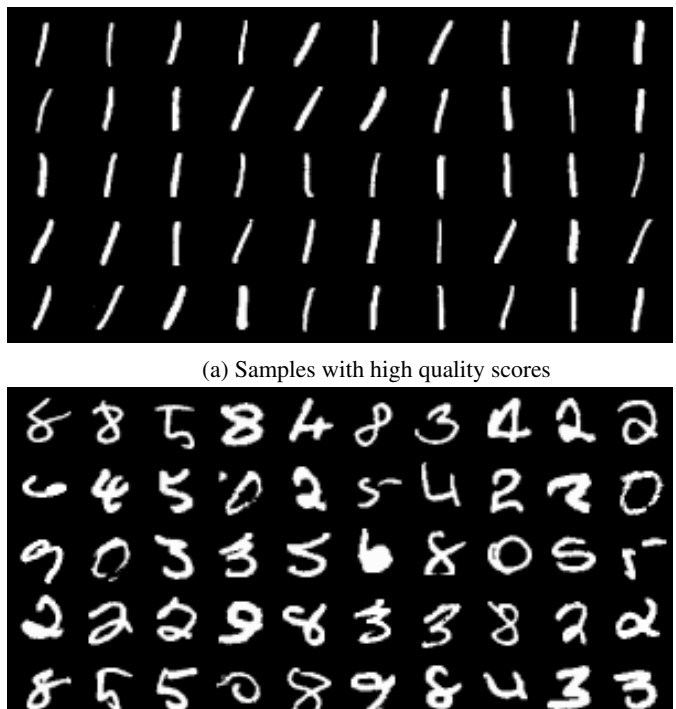

(a) Samples with high quality scores

(b) Samples with low quality scores

Figure 14: Sample quality check by iWGAN on the validation dataset of MNIST

## APPENDIX H: MORE EXPERIMENTAL RESULTS ON CELEBA

### H.1 LATENT SPACE

Figure 15 shows the first 8 dimensions of the latent space calculated by $Q(x)$ on CelebA.

### H.2 RANDOM GENERATED SAMPLE

Figure 16, Figure 17, Figure 18 and Figure 19 display the random generated samples from Wasserstein GAN gradient penalty(WGAN-GP), iWGAN, Wasserstein Autoencoder(WAE), and Adversarial Learning Inference (ALI), respectively. Table 1 shows numerical comparison among these four methods based on inception scores (IS), Fréchet inception distances (FID), reconstruction errors (RE), and maximum mean discrepancy (MMD) between encodings and standard normal random variables.

The IS and FID scores are calculated based on pre-trained inception models. However, as discussed in Barratt & Sharma (2018), inception score is not a reliable metric for the wellness of generated samples. This is also consistent with our experiments. Although WAE delivers the best inception scores among four methods, WAE also has the worst FID scores. The generated samples (Figure 18) show that WAE is not the best generative model compared with other three methods.

The reconstruction error (RE) is defined as

$$RE = \frac{1}{N} \sum_{i=1}^{N} \|\hat{X}_i - X_i\|_2, \tag{15}$$

$\hat{X}_i$ is the reconstructed sample for $X_i$. RE is used to measure if the method has generated meaningful latent encodings. Smaller reconstruction errors indicate a more meaningful latent space which can be decoded into the original samples.

The maximum mean discrepancy (MMD) is defined as

$$MMD = \frac{1}{N(N-1)} \sum_{l \neq j} k(z_l, z_j) + \frac{1}{N(N-1)} \sum_{l \neq j} k(\tilde{z}_l, \tilde{z}_j) - \frac{2}{N^2} \sum_{l,j} k(z_l, \tilde{z}_j) \tag{16}$$

where $k$ is a positive-definite reproducing kernel, $z_i$s are drawn from prior distribution $\mathbb{P}_Z$, and $\tilde{z}_i = Q(x_i)$ are the latent encodings of real samples. MMD is used to measure the difference between distribution of latent encodings and standard normal random variables. Smaller MMD indicates that the distribution of encodings is close to the standard normal distribution.

From Table 1, in terms of generative models, iWGAN and ALI are better models, where WGAN-GP comes after, but WAE is suffering from generating clear pictures. In terms of RE and MMD, iWGAN and WAE are better choices, where ALI cannot always reconstruct the sample to itself (Figure 21). In general, Table 1 shows that iWGAN has successfully produce both meaningful encodings and reliable generator simultaneously.

Table 1: Comparison of iWGAN, ALI, WAE, WGAN-GP

| Methods | IS | FID | RE | MMD |
|---------|-----|------|------|------|
| True | 1.96(0.019) | 18.63 | – | – |
| iWGAN | 1.51(0.017) | **51.20** | **13.55(2.41)** | $6 \times 10^{-3}$ |
| ALI | 1.50(0.014) | **51.12** | 34.49(8.23) | 0.39 |
| WAE | **1.71(0.029)** | 77.53 | **9.88(1.42)** | $4 \times 10^{-3}$ |
| WGAN-GP | **1.54(0.016)** | 61.39 | – | – |

Figure 20 shows the interpolation comparison between iWGAN, WAE and ALI.

### H.3: RECONSTRUCTION

Figure 22 shows the distribution of reconstruction error of CelebA. Figure 21 shows the comparison between real images and reconstructed images among three methods, iWGAN, WAE and ALI. Figure 23 shows samples with high and low quality scores in CelebA validation sets.

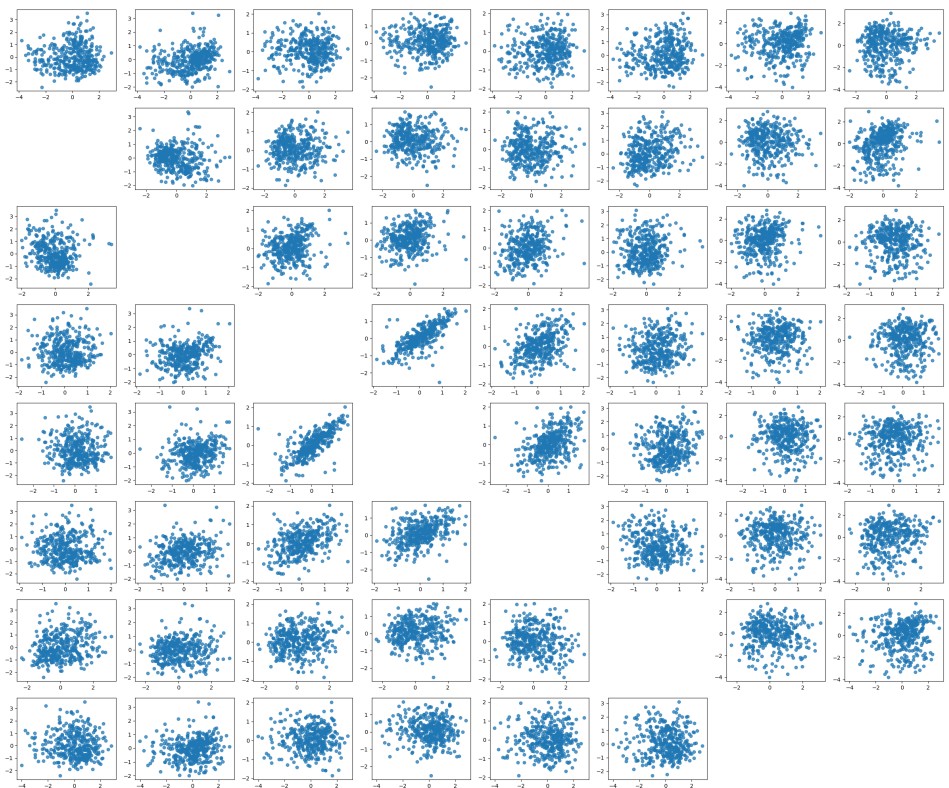

Figure 15: Latent Space of CelebA dataset

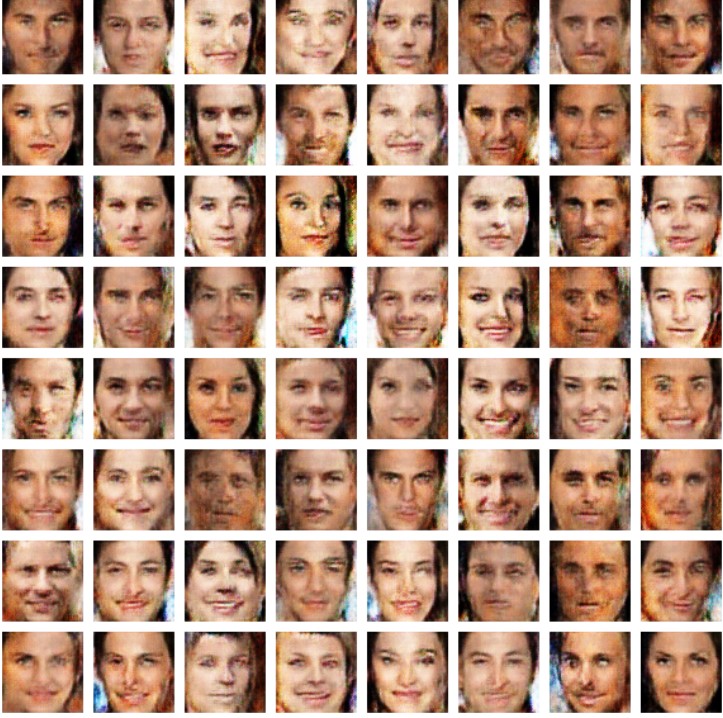

Figure 16: Generated samples by WGAN-GP

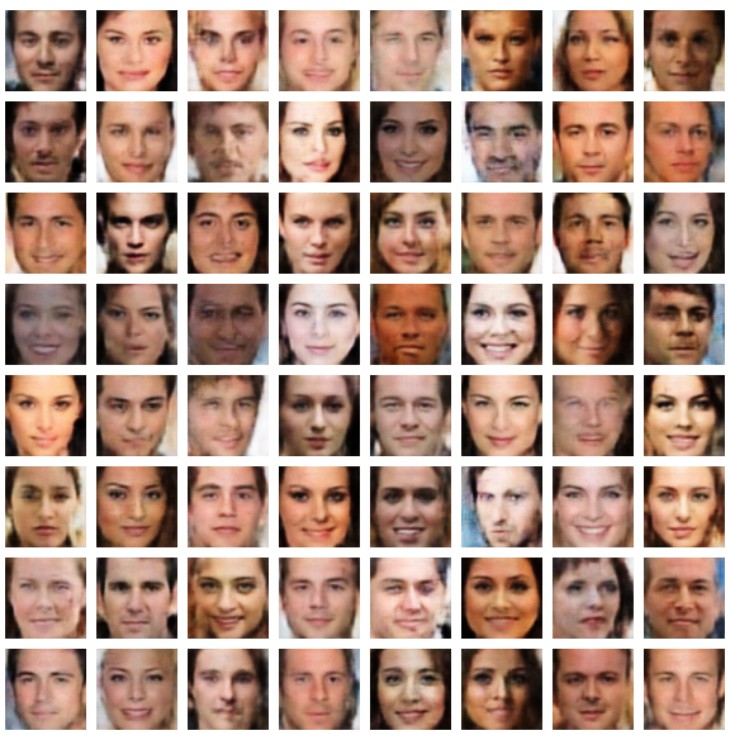

Figure 17: Generated samples by iWGAN

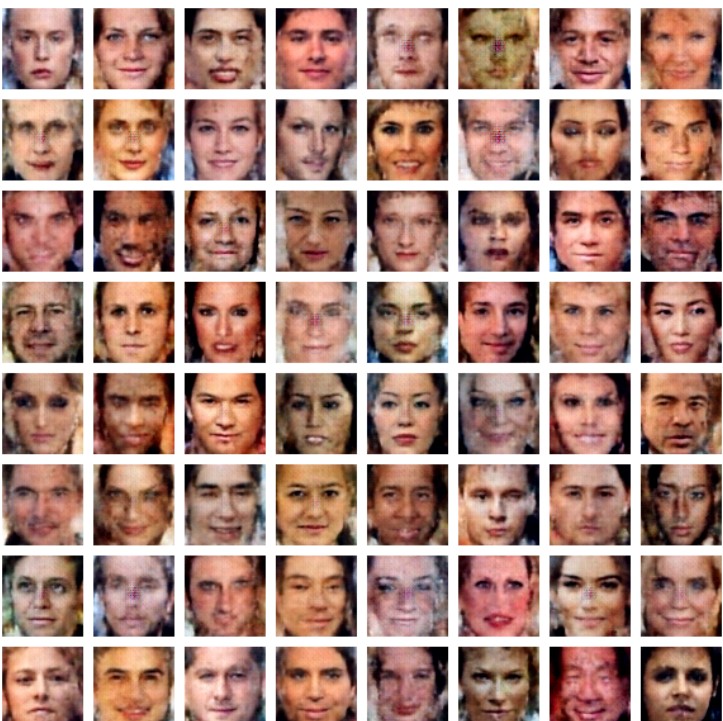

Figure 18: Generated samples by WAE

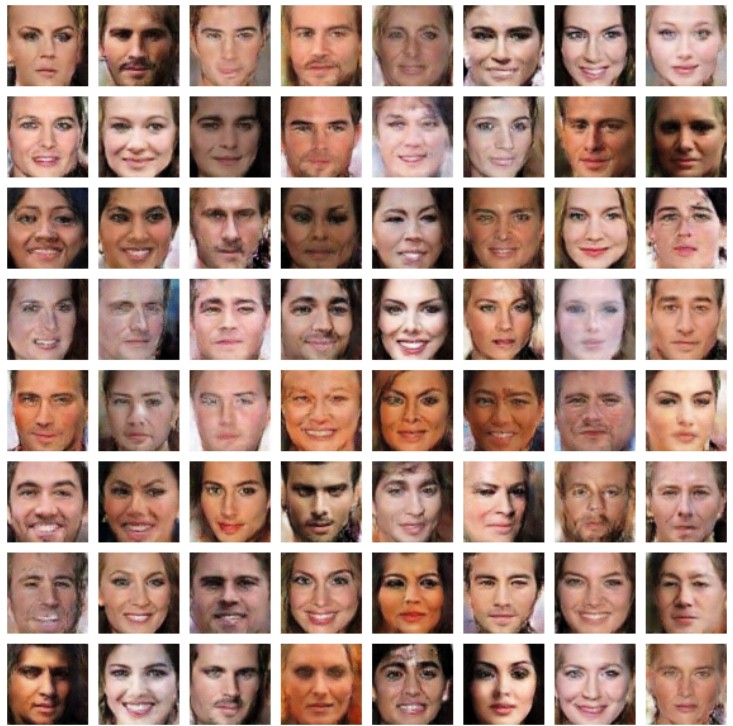

Figure 19: Generated samples by ALI

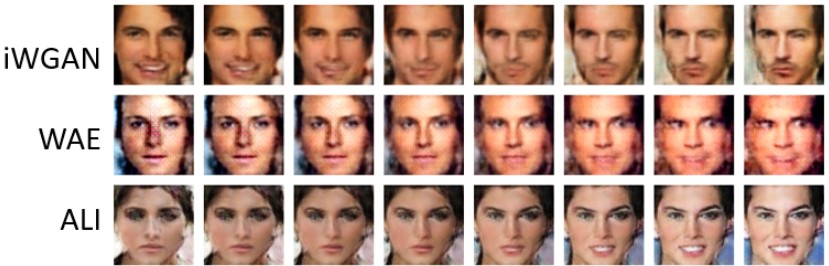

Figure 20: Interpolations comparison among iWGAN, WAE and ALI

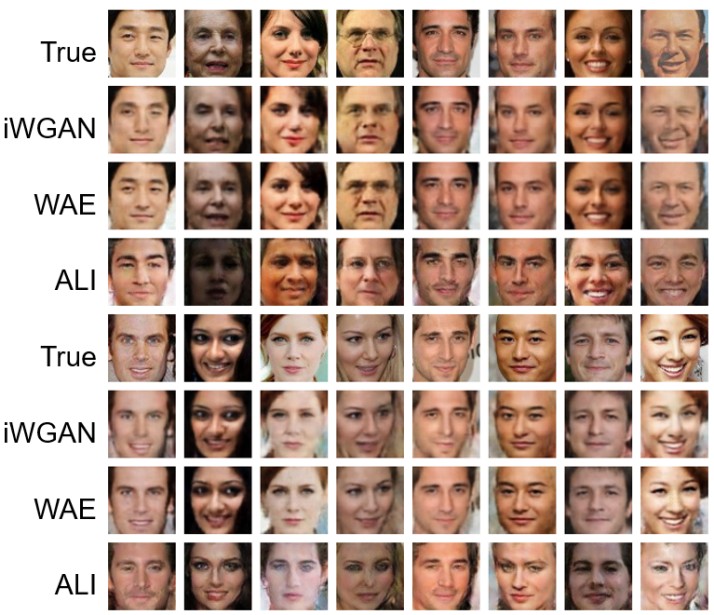

Figure 21: Reconstructed samples comparison among iWGAN, WAE and ALI

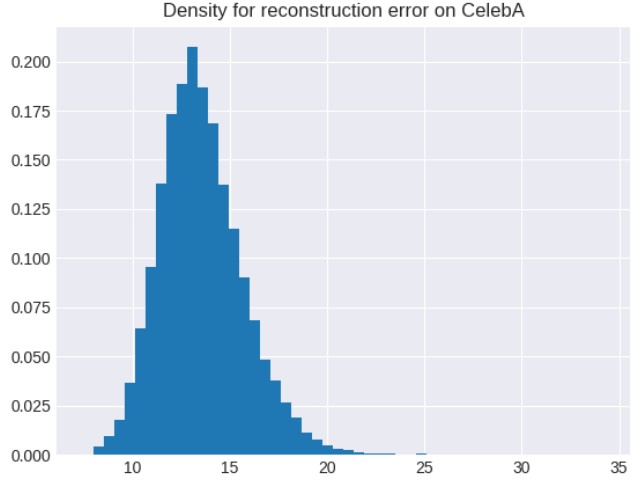

Figure 22: Histogram of reconstruction errors on CelebA

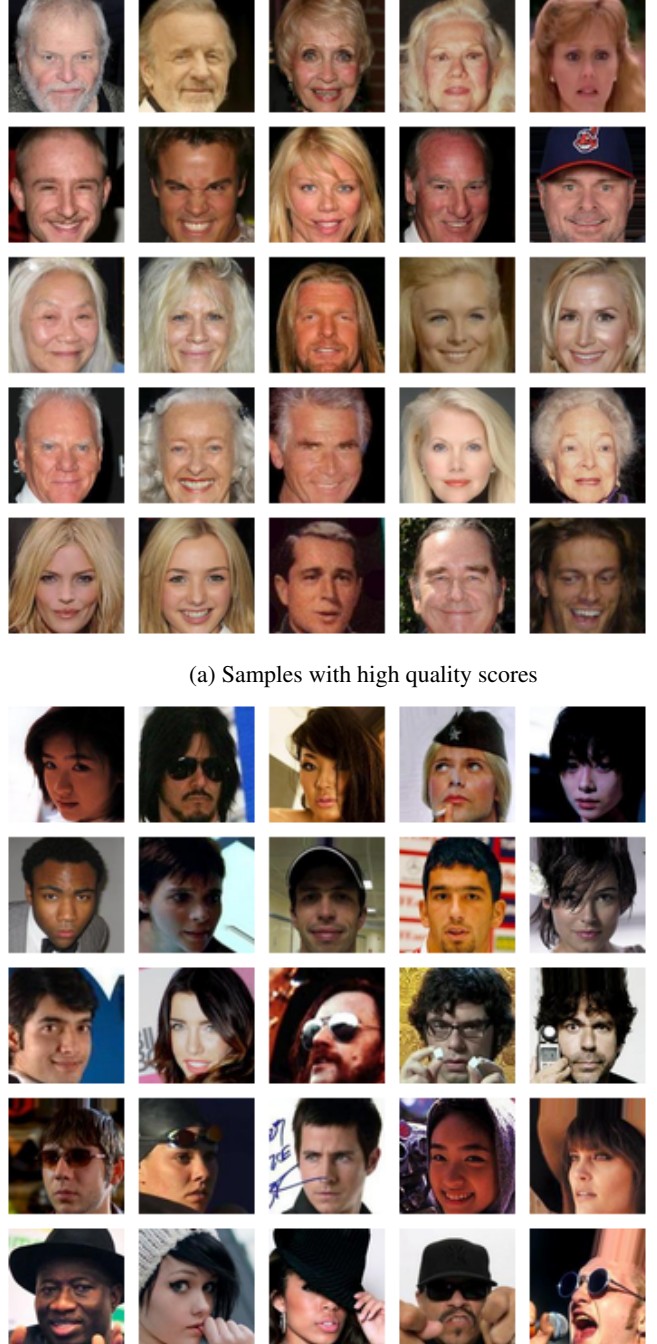

(a) Samples with high quality scores

(b) Samples with lower quality scores

Figure 23: Sample quality check by iWGAN on CelebA

