# OpenReview forum: "iWGAN: an Autoencoder WGAN for Inference"
_ICLR.cc/2020/Conference — Reject_

### Official Review · AnonReviewer3 · 2019-10-22
**Official Blind Review #3**

**Rating:** 8

**Review:**

This paper presents an inference WGAN (iWGAN) which fully considers to reduce the difference between distributions of G(X) and Z, G(Z) and X. In this algorithm, the authors show a rigorous probabilistic interpretation under the maximum likelihood principle. This algorithm has a stable and efficient training process. The authors provided a lots of theoretical and experimental analysis to show the effectiveness of the proposed algorithm. Therefore, the innovation of this paper is very novel. The theoretical analysis is sufficient, and the technology is sound.

**Experience Assessment:**

I have read many papers in this area.

**Review Assessment: Checking Correctness Of Derivations And Theory:**

I did not assess the derivations or theory.

**Review Assessment: Checking Correctness Of Experiments:**

I assessed the sensibility of the experiments.

**Review Assessment: Thoroughness In Paper Reading:**

I read the paper at least twice and used my best judgement in assessing the paper.

---

> ### Author Response · Authors · 2019-11-11
> **Thank you for your encouragement!**
>
> We thank the reviewer for the valuable comments and encouragement.
>
> --------------------------------
> Comment 1: This paper presents an inference WGAN (iWGAN) which fully considers to reduce the difference between distributions of G(X) and Z, G(Z) and X. In this algorithm, the authors show a rigorous probabilistic interpretation under the maximum likelihood principle. This algorithm has a stable and efficient training process. The authors provided a lots of theoretical and experimental analysis to show the effectiveness of the proposed algorithm. Therefore, the innovation of this paper is very novel. The theoretical analysis is sufficient, and the technology is sound.
>
> Response 1: Thank you very much for your encouragement and positive comments.

---

### Official Review · AnonReviewer2 · 2019-10-23
**Official Blind Review #2**

**Rating:** 3

**Review:**

In this paper, an inference WGAN model (iWGAN) is proposed that fuses autoencoders and WGANs. By working on both the prime and dual problems, the proposed iWGAN takes an objective that is in general the upper bound of WGAN. The generalization error bound of iWGAN is analysed and its probabilistic interpretation under maximum likelihood estimation is provided.  The proposed iWGAN model is validated on both synthetic and real (MNIST and CelebA) datasets.

Pros:
This paper not only proposed the iWGAN model, but also provided some theoretical analysis about the iWGAN, such as the generalization error bound and the probabilistic interpretation.
Through the experiment, it seems that the proposed iWGAN works.  Especially, on the synthetic dataset, iWGAN seems to  have less mode collapse than WGAN.

Cons:
The major concerns of this paper lie in its experiment.
(1)	There is no quantitative comparison between iWGAN and WGAN, especially on the two real image datasets.  There are only some visual examples of the results in either the main body of the text or the appendix.
(2)	The proposed iWAN is only compared with WGAN.  There is no experimental comparison with other autoencoder GANs  in the literature,  although the paper stated that iWGAN has many advantages over other autoencoder GANs (in the abstract).

In addition to experiment, other concerns include:
(1)	The proposed method needs to be better motivated, although it is interpreted under the framework of maximum likelihood estimation. For example, what is the advantage to optimize the proposed upper bound of WGAN over its original objective? In the experiment, it seems iWGAN has less mode collapse than WGAN based on the synthetic data. But there is no explanation or discussion about why iWGAN could enjoy this favourable property.
(2)	The proposed framework is tightly based on WGAN. It is not clear whether it could be extended to other GANs, which limits the significance of the proposed work.

Due to the above concerns, rating is recommended as "3 Weak Reject."

Suggestions:  The authors are encouraged to provide extensive comparison with other autoencoder GANs and WGAN, especially in quantitative way.


**Experience Assessment:**

I have published one or two papers in this area.

**Review Assessment: Checking Correctness Of Derivations And Theory:**

I assessed the sensibility of the derivations and theory.

**Review Assessment: Checking Correctness Of Experiments:**

I carefully checked the experiments.

**Review Assessment: Thoroughness In Paper Reading:**

I read the paper at least twice and used my best judgement in assessing the paper.

---

> ### Author Response · Authors · 2019-11-11
> **Thank you for your suggestions!**
>
> We thank the reviewer for the valuable suggestions and comments. We have addressed your concerns in the updated revision and the following response:
>
> --------------------------------
> Comment 1: There is no quantitative comparison between iWGAN and WGAN, especially on the two real image datasets.  There are only some visual examples of the results in either the main body of the text or the appendix.
>
> Response 1: Thank you very much for your suggestion. For CelebA, we have performed more experiments and compared iWGAN with WGAN-GP, Adversarial Learning Inference (ALI) and Wasserstein Autoencoder (WAE) in terms of four performance measures such as Inception Scores, FID scores, Reconstruction errors, and Maximum Mean Discrepancy between latent encodings and standard normal variables. The results are displayed in Table 1 in Appendix H. These results demonstrate that iWGAN has a competitive performance for all these three tasks simultaneously. Note that pre-trained inception models for both Inception scores and FID scores are based on RGB images, i.e. 3-channel inputs, so that it is not suitable for grey-scale images like MNIST. This is why we do not include the quantiative results for MNIST.
>
> --------------------------------
> Comment 2: The proposed iWAN is only compared with WGAN.  There is no experimental comparison with other autoencoder GANs  in the literature,  although the paper stated that iWGAN has many advantages over other autoencoder GANs (in the abstract).
>
> Response 2: Thank you very much for your suggestion. For CelebA, we have added both quantitative comparison and visual comparison between iWGAN and two autoencoder GANs: Adversarial Learning Inference (ALI) and Wasserstein Autoencoder (WAE). The results are displayed in Appendix H.
>
> --------------------------------
> Comment 3: The proposed method needs to be better motivated, although it is interpreted under the framework of maximum likelihood estimation. For example, what is the advantage to optimize the proposed upper bound of WGAN over its original objective? In the experiment, it seems iWGAN has less mode collapse than WGAN based on the synthetic data. But there is no explanation or discussion about why iWGAN could enjoy this favourable property.
>
> Response 3: Thank you very much for your suggestion. We have re-written Section 2 to make the motivations clear. The advantages of iWGAN objective are:
>     -- Our goal is to propose an autoencoder generative model which satisfies the following three conditions $simultaneously$: (a) The generator can generate images which have a similar distribution with observed images; (b) The encoder can produce meaningful encodings in the latent space; (c) The reconstruction errors of this model based on these meaningful encodings are small.
>     -- We have provided the weakness of both WGAN and WAE. Specifically, WGAN does not produce meaningful encodings and many experiments still display the problem of mode collapse. WAE defines a generative model in an implicit way and does not model the generator through $G(Z)$ with $Z\sim P_Z$ directly.
>     -- The proposed model, iWGAN, is able to take the advantages if both WGAN and WAE. The objective function of iWGAN is a tight upper bound of the WGAN objective.
>
> The reason why iWGAN can avoid mode-collapse is because that using an autoencoder is to encourage the model to better represent $all$ data it is trained with, so that it discourages mode collapse.
>
> --------------------------------
> Comment 4: The proposed framework is tightly based on WGAN. It is not clear whether it could be extended to other GANs, which limits the significance of the proposed work.
>
> Response 4: It is straightforward to extend the iWGAN framework to other GANs. The extension to $f$-GANs is given in Appendix D.

---

### Official Review · AnonReviewer1 · 2019-10-29
**Official Blind Review #1**

**Rating:** 3

**Review:**

This paper proposes a novel variation to the WGAN, which combines WGANs with autoencoders. The paper contains several interesting ideas and theoretical results. The proposed method is also demonstrated to perform reasonable in benchmark examples. One particularly nice property is that the authors derive a duality gap that can be used as a stopping criterion.

My main concern with the paper is that I don’t think it is sufficiently well written. In particular, the motivations for IWGAN presented in section 2 are both technically involved and vague. For instance, the optimisation problem is introduced with the sentence “The primal and dual formulations motivate us to define the iWGAN objective to be”, which does not motivate what we might benefit from using the objective. I generally feel that the presented arguments in favour of IWGAN are not sufficiently clear and convincing. Unfortunately, I also have similar concerns with later sections. In fact, even though the authors argue that the simulation results are also better than state of the art algorithms, my impression is that recent GAN modules often generate images which are even more realistic than this.

In spite of my concerns, I find the presented theory interesting and with better motivations and examples, it may eventually become a solid and well-cited contribution.

**Experience Assessment:**

I have read many papers in this area.

**Review Assessment: Checking Correctness Of Derivations And Theory:**

I assessed the sensibility of the derivations and theory.

**Review Assessment: Checking Correctness Of Experiments:**

I did not assess the experiments.

**Review Assessment: Thoroughness In Paper Reading:**

I read the paper at least twice and used my best judgement in assessing the paper.

---

> ### Author Response · Authors · 2019-11-11
> **Thank you for your review!**
>
> We thank the reviewer for valuable suggestions and comments. We have addressed your concerns in the updated revision and the following response:
>
> --------------------------------
> Comment 1: I don’t think it is sufficiently well written. In particular, the motivations for IWGAN presented in section 2 are both technically involved and vague. For instance, the optimisation problem is introduced with the sentence “The primal and dual formulations motivate us to define the iWGAN objective to be”, which does not motivate what we might benefit from using the objective. I generally feel that the presented arguments in favour of IWGAN are not sufficiently clear and convincing.
>
> Response 1: Thank you very much for your suggestion. We have re-written Section 2 to make the motivations clear. The motivations for iWGAN can be summarized as follows:
>     -- The benefit of using an autoencoder is to encourage the model to better represent $all$ data it is trained with, so that it discourages mode collapse.
>     -- Our goal is to propose an autoencoder generative model which satisfies the following three conditions $simultaneously$: (a) The generator can generate images which have a similar distribution with observed images; (b) The encoder can produce meaningful encodings in the latent space; (c) The reconstruction errors of this model based on these meaningful encodings are small.
>     -- We have provided the weakness of both WGAN and WAE. Specifically, WGAN does not produce meaningful encodings and many experiments still display the problem of mode collapse. WAE defines a generative model in an implicit way and does not model the generator through $G(Z)$ with $Z\sim P_Z$ directly.
>     -- The proposed model, iWGAN, is able to take the advantages if both WGAN and WAE. The objective function of iWGAN is a tight upper bound of the WGAN objective.
>     -- The extension of our framework to other GANs is straightforward, and the extension to $f$-GAN is given in Appendix D.
>
> --------------------------------
> Comment 2: In fact, even though the authors argue that the simulation results are also better than state of the art algorithms, my impression is that recent GAN modules often generate images which are even more realistic than this.
>
> Response 2: Thank you very much for your suggestion. We agree with you that the recent GAN modules such as BigGAN can produce high-resolution and high-fidelity images. As its name suggests, the BigGAN focuses on scaling up the GAN models including more model parameters, larger batch sizes, and architectural changes.
> Instead, we propose a new framework which is able to complete the above three tasks simultaneously, i.e., a good generative model, meaningful encodings, and small reconstruction errors. We have added more experiments and compared iWGAN with WGAN-GP, Adversarial Learning Inference (ALI) and Wasserstein Autoencoder (WAE) in terms of Inception Scores, FID scores, Reconstruction errors, and Maximum Mean Discrepancy between latent encodings and standard normal variables in Appendix H. All experiments are done with similar settings, including model structures, number of parameters, hyper-parameters etc. These results demonstrate that iWGAN has a competitive performance for all these three tasks simultaneously. Given enough computing resources, iWGAN has the ability to generate high-resolution and high-quality images.
>
> --------------------------------
> Comment 3: In spite of my concerns, I find the presented theory interesting and with better motivations and examples, it may eventually become a solid and well-cited contribution.
>
> Response 3: Thank you very much for your encouragement. We hope our revision has addressed your concerns.

---

### Decision · Program_Chairs · 2019-12-19

**Decision:**

Reject

**Comment:**

This paper proposes a new way to stabilise GAN training.

The reviews were very mixed but taken together below acceptance threshold.

Rejection is recommended with strong motivation to work on the paper for next conference. This is potentially an important contribution.